# Ocean acidification and high irradiance stimulate the photophysiological fitness, growth and carbon production of the Antarctic cryptophyte *Geminigera cryophila*

Scarlett Trimborn[1,2], Silke Thoms[1], Pascal Karitter[1], Kai Bischof[2]

[1]EcoTrace, Biogeosciences section, Alfred Wegener Institute, Bremerhaven, 27568, Germany
[2]Marine Botany, Department 2 Biology/Chemistry, University of Bremen, Bremen, 28359, Germany

*Correspondence to*: Scarlett Trimborn (scarlett.trimborn@awi.de)

**Abstract.**

Ecophysiological studies on Antarctic cryptophytes to assess whether climatic changes such as ocean acidification and enhanced stratification affect their growth in Antarctic coastal waters in the future are lacking so far. This is the first study that investigated the combined effects of increasing availability of $pCO_2$ (400 and 1000 µatm) and irradiance (20, 200 and 500 µmol photons $m^{-2}$ $s^{-1}$) on growth, elemental composition and photophysiology of the Antarctic cryptophyte *Geminigera cryophila*. Under ambient $pCO_2$, this species was characterized by a pronounced sensitivity to increasing irradiance with complete growth inhibition at the highest light intensity. Interestingly, when grown under high $pCO_2$ this negative light effect vanished and it reached highest rates of growth and particulate organic carbon production at the highest irradiance compared to the other tested experimental conditions. Our results for *G. cryophila* reveal beneficial effects of ocean acidification in conjunction with enhanced irradiance on growth and photosynthesis. Hence, cryptophytes such as *G. cryophila* may be potential winners of climate change, potentially thriving better in more stratified and acidic coastal waters and contributing in higher abundance to future phytoplankton assemblages of coastal Antarctic waters.

## 1 Introduction

Even though Antarctic coastal waters comprise a relatively small area relative to the open ocean, these waters are highly productive due to the constant supply of macronutrients and iron (Arrigo et al., 2008). Shelf waters adjacent to the Western Antarctic Peninsula (WAP) are currently undergoing rapid physical changes, exhibiting the most rapid warming rates than anywhere in Antarctica over the last decades (Ducklow et al., 2007, 2013). Rising air temperature resulted in shorter sea ice seasons (Smith and Stammerjohn, 2001) with contrasting effects on phytoplankton biomass, composition and productivity between the northern and southern WAP. For the latter, the earlier retreat of sea ice together with the observed increase in surface water temperature led to shallow water column stratification, which favored phytoplankton growth and productivity. In the northern part of the WAP on the other hand, the earlier disappearance of sea ice was associated to greater wind activities and more cloud formation. As a consequence, a deepening of the upper mixed layer was found, providing less favorable light conditions. Next to reduced chlorophyll *a* accumulation (Montes-Hugo et al., 2009) and primary production (Moreau et al., 2015), a decline of large phytoplankton such as diatoms relative to the whole community was observed (Montes-Hugo et al., 2009; Rozema et al., 2017). Accordingly, a recurrent shift from diatoms to cryptophytes and small flagellates was reported for waters north of the WAP, with important implications for food web dynamics (Moline et al., 2004; Ducklow et al., 2007; Montes-Hugo et al., 2009; Mendes et al., 2017). The frequent occurrence of cryptophytes was previously reported after diatom blooms (Moline and Prezelin, 1996) and was related to surface melt water

stratification (Moline et al., 2004). As a result from rising global air temperatures, surface water freshening is expected to shallow the upper water layer, exposing phytoplankton to higher light intensity (Moreau et al., 2015). Relative to diatoms and the prymnesiophyte *Phaeocystis antarctica*, cryptophytes were found to be the main contributors to biomass in stratified and warm WAP coastal waters potentially resulting from their high tolerance to withstand high irradiances (Mendes et al., 2017). Considering the lack in ecophysiological studies carried out with Antarctic ecologically relevant cryptopyte species it remains yet unclear whether the projected climatic changes could promote cryptophyte growth in Antarctic coastal waters in the future. Hence, higher abundances of cryptophytes could have important implications for the biogeochemistry of these waters, as they are considered to be inefficient vectors of carbon and thus could reduce the efficiency of the biological carbon pump.

Light availability strongly influences the rate of growth and carbon fixation of phytoplankton (Falkowski and Raven 2007). With increasing irradiance, Antarctic phytoplankton species exhibited increased growth and carbon fixation, but only until photosynthesis was saturated (Fiala and Oriol, 1990; Heiden et al., 2016). When exposed to excessive radiation, phytoplankton cells can get photoinhibited and even damaged (Falkowski and Raven, 2007). Cryptophytes are exceptional among eukaryotic microalgae as they contain similar to diatoms chlorophyll a/c proteins, the carotenoid alloxanthin and phycobiliproteins homologous to red algal phycobiliproteins (Gould et al., 2008). This pigment composition allows crypotophytes to cope well under limited irradiance. Different to diatoms and prymnesiophytes, cryptophytes have no photoprotective de-epoxidation/epoxidation cycling of xanthophyll pigments (e.g. diadinoxanthin, diatoxanthin), instead they rely on chlorophyll a/c proteins, which function to dissipate excess light energy, as another type of nonphotochemical quenching (NPQ; Funk et al., 2011; Kana et al., 2012). While laboratory studies so far mainly have concentrated to disentangle the physiological response of Southern Ocean key species of diatom and prymnesiophytes to different environmental factors almost nothing is known on Antarctic cryptophytes. Apart from field studies (Moline et al., 2004; Ducklow et al., 2007; Montes-Hugo et al., 2009), almost nothing is known on how climate change could influence the ecophysiology of Antarctic cryptophytes. Due to the increased solubility of $CO_2$ in cold water, ocean acidification (OA) is anticipated to strongly affect polar waters (Orr et al., 2005; IPCC Report, 2014). For various Antarctic diatoms and the prymnesiophyte *P. antarctica*, growth and/or carbon fixation remained unaltered by OA alone (Riebesell et al., 1993; Boelen et al., 2011; Hoogstraten et al., 2012; Trimborn et al., 2013; Hoppe et al., 2015; Heiden et al., 2016). Recent studies suggest that Southern Ocean diatoms are more prone to OA especially in conjunction with high light than the prymnesiophyte *Phaeocystis antarctica* (Feng et al., 2010; Trimborn et al., 2017a,b; Beszteri et al. 2018; Heiden et al., 2018; Koch et al., 2018, Heiden et al., 2019). The response of cryptophytes to OA is hitherto almost unexplored. The few studies on temperate phytoplankton communities suggest that cryptophytes were not affected by OA (Schulz et al., 2017), even when exposed in combination with increased ultraviolet radiation (Domingues et al., 2014) or warming (Sommer et al., 2015). Similarly, no discernible OA effect on cryptophyte abundance was found in subantarctic (Donahue et al., 2019) and Antarctic (Young et al., 2015) natural phytoplankton communities composed of diatoms, cryptophytes, and *Phaeocystis* spp..

Based on previous studies on the single effects of light or $CO_2$ alone, we hypothesize that cryptophytes are able to cope well with OA and high light conditions. Due to the limited information available on Antarctic cryptophyte physiology, this study assessed OA effects and their interaction with increasing irradiance on the physiology of the Antarctic cryptophyte *Geminigera cryophila*. To this end, *G. cryophila* was grown under two

$pCO_2$ levels (400 and 1000 µatm) in combination with three irradiance levels (20, 200 and 500 $\mu$mol photons m$^{-2}$ s$^{-1}$) and their interactive effects on growth, elemental composition and photophysiology were assessed.

## 2 Material and Methods

### 2.1 Culture conditions

Triplicate semi-continuous cultures of the Antarctic cryptophyte *Geminigera cryophila* (CCMP 2564) were grown in exponential phase at 2 °C in sterile-filtered (0.2 $\mu$m) Antarctic seawater (salinity 30.03). This seawater was enriched with phosphate (final concentration of 100 $\mu$mol L$^{-1}$), nitrate (final concentration of 6.25 $\mu$mol L$^{-1}$) (N:P ratio of 16:1, Redfield, 1963) as well as trace metals and vitamins according to F/2 medium (Guillard and Ryther, 1962). *G. cryophila* cells were grown under a 16h light: 8h dark cycle at three constant light intensities (LL = 20, ML = 200 and HL = 500 $\mu$mol photons m$^{-2}$ s$^{-1}$) using light-emitting diodes (LED) lamps (SolarStinger LED SunStrip Marine Daylight, Econlux). Light intensities were adjusted using a LI-1400 datalogger (Li-Cor, Lincoln, NE, USA) with a 4π-sensor (Walz, Effeltrich, Germany). The three light treatments were further combined with two $CO_2$ partial pressures ($pCO_2$) of 400 (ambient $pCO_2$ treatment) or 1000 $\mu$atm (OA treatment, Table 1). All $pCO_2$ treatments and the respective dilution media were continuously and gently bubbled through a frit with humidified air of the two $pCO_2$ levels, which were generated from $CO_2$-free air (< 1 ppmv $CO_2$; Dominick Hunter, Kaarst, Germany) and pure $CO_2$ (Air Liquide Deutschland ltd., Düsseldorf, Germany) with a gas flow controller (CGM 2000, MCZ Umwelttechnik, Bad Nauheim, Germany). The $CO_2$ gas mixtures were regularly monitored with a nondispersive infrared analyzer system (LI6252; Li-Cor Biosciences) calibrated with $CO_2$-free air and purchased gas mixtures of 150 ± 10 and 1000 ± 20 ppmv $CO_2$ (Air Liquide Deutschland). Dilutions with the corresponding acclimation media ensured that the pH level remained constant (±0.07 units, Table 1) and that the cells stayed in the exponential growth phase. *G. cryophila* cells were acclimated to the matrix of three light intensities (LL = 20, ML = 200 and HL = 500 $\mu$mol photons m$^{-2}$ s$^{-1}$) and two $pCO_2$ levels (ambient = 400 and OA = 1000 µatm) for at least two weeks prior to the start of the main experiment. Despite several attempts, *G. cryophila* did not grow at ambient $pCO_2$ in conjunction with HL. For the main experiments, cells grew in exponential phase and final sampling took place between 7,280 and 17,161 cells per mL.

### 2.1 Seawater carbonate chemistry

The pH of the different cultures and the culture medium was measured every other day and at the final sampling day using a pH-ion meter (pH-Meter 827, Metrohm), calibrated (3 point calibration) with National Institute of Standards and Technology-certified buffer systems. The pH remained constant at 8.13 ± 0.07 and 7.82 ± 0.06 for the ambient $pCO_2$ and OA treatments, respectively (Table 1). Dissolved inorganic carbon (DIC) samples were sterile-filtered (0.2 $\mu$m) and stored at 4 °C in 5 mL gas-tight borosilicate bottles without headspace until analysis. DIC was measured colourimetrically in duplicates with a QuAAtro autoanalyzer (Seal Analytical, Stoll et al., 2001). The carbonate system was calculated based on DIC, pH, silicate (97 µmol kg$^{-1}$), phosphate (6.1 µmol kg$^{-1}$), temperature (2.0 °C) and salinity (30.03) using the CO2Sys program (results shown in Table 1, (Pierrot et al., 2006) choosing the equilibrium constant of Mehrbach et al. (1973) refitted by Dickson and Millero (1987).

### 2.2 Growth, elemental stoichiometry and composition

Cell count samples of every *G. cryophila* treatment were taken on a daily basis at the same time of day and were determined immediately after sampling using a Coulter Multisizer III (Beckmann-Coulter, Fullerton, USA). Cell-specific growth rate ($\mu$, unit d$^{-1}$) was calculated as

$$\mu = (\ln N_{\text{fin}} - \ln N_0)/\Delta t, \tag{1}$$

where $N_0$ and $N_{\text{fin}}$ denote the cell concentrations at the beginning and the end of the experiments, respectively, and $\Delta t$ is the corresponding duration of incubation in days.

Particulate organic carbon (POC) and particulate organic nitrogen (PON) were measured after filtration onto precombusted (12 h, 500 °C) glassfibre filters (GF/F, pore size ~0.6 $\mu$m, Whatman). Filters were stored at -20 °C and dried for >12 h at 64 °C prior to sample preparation. Analysis was performed using an Euro Vector CHNS-O elemental analyzer (Euro Elemental Analyzer 3000, HEKAtech GmbH, Wegberg, Germany). Contents of POC and PON were corrected for blank measurements and normalized to filtered volume and cell densities to yield cellular quotas. Production rates of POC and PON were calculated by multiplication of the cellular quota with the specific growth rate of the respective treatment.

**2.3 Chlorophyll *a* fluorescence**

The efficiency of photochemistry in photosystem II (PSII) was assessed in all treatments using a Fast Repetition Rate fluorometer (FRRf, FastOcean PTX; Chelsea Technologies Group ltd., West Molesey, United Kingdom) in combination with a FastAct Laboratory system (Chelsea Technologies Group ltd., West Molesey, United Kingdom). Cells of the respective treatment were dark-acclimated for 10 min, before minimum chlorophyll *a* (Chl *a*) fluorescence ($F_o$) was recorded. Subsequently, a single turnover flashlet (1.2 x 10$^{22}$ photons m$^{-2}$ s$^{-1}$, wavelength 450 nm) was applied to cumulatively saturate photosystem II (PSII), i.e. a single photochemical turnover (Kolber et al., 1998). The single turnover saturation phase comprised 100 flashlets on a 2 $\mu$s pitch and was followed by a relaxation phase comprising 40 flashlets on a 50 $\mu$s pitch. This sequence was repeated 24 times within each acquisition. The saturation phase of the single turnover acquisition was fitted according to Kolber et al. (1998). From this measurement, the minimum ($F_o$) and maximum ($F_m$) Chl *a* fluorescence was determined. Using these two parameters, the dark-adapted maximum PSII quantum yield ($F_v/F_m$) was calculated according to the equation ($F_m - F_o$)$/F_m$. During the fluorescence light curve (FLC), cells were exposed for 5 min to eight actinic light levels ranging from 35 to 1324 $\mu$mol photons m$^{-2}$ s$^{-1}$. From these measurements, the light-adapted minimum ($F'$) and maximum ($F_m'$) fluorescence of the single turnover acquisition was estimated. The effective PSII quantum yield under ambient light ($F_q'/F_m'$) was derived according to the equation ($F_m' - F'$)$/F_m'$ (Genty et al., 1989). From this curve, absolute electron transport rates (ETR, e$^{-}$ PSII$^{-1}$ s$^{-1}$) were calculated following Suggett et al. (2004, 2009):

$$\text{ETR} = \sigma_{\text{PSII}} \text{ x } (F_q'/F_m' \, / \, F_v/F_m) \text{ x E}, \tag{2}$$

where $\sigma_{\text{PSII}}$ is the functional absorption cross section of PSII photochemistry and E denotes the applied instantaneous irradiance (photons m$^{-2}$ s$^{-1}$). Light-use characteristics were analyzed by fitting irradiance-dependent ETRs according to Ralph and Gademann (2005), including maximum ETR (ETR$_{\text{max}}$), minimum saturating irradiance ($I_K$) and maximum light utilization efficiency ($\alpha$). Using the Stern-Volmer equation, nonphotochemical quenching (NPQ) of chlorophyll *a* fluorescence was calculated as $F_m \, / \, F_m' - 1$. From the single turnover measurement of dark-adapted cells, $\sigma_{\text{PSII}}$, the energy transfer between PSII units (i.e. connectivity, *P*), the re-oxidation of the electron acceptor Q$_a$ ($\tau$) and the concentration of functional photosystem II reaction centers ([RII]) were assessed from iterative

algorithms for induction (Kolber et al., 1998) and relaxation phase (Oxborough, 2012). [RII] represents an estimator for the content of PSII in the sample and was calculated according to the following equation:

$$[RII] = (F_o / \sigma_{PSII}) \times (K_R / E_{LED}),\qquad\qquad(3)$$

where $K_R$ is an instrument specific constant and $E_{LED}$ is the photon flux from the FRRf measuring LEDs. After the completion of the FLC curve, an additional dark-adaptation period of 10 min was applied, followed by a single turnover flashlet to check for recovery of PSII. Using the $F_v/F_m$ measured before and after the FLC-curve, the yield recovery was calculated and given as % of the initial $F_v/F_m$ (before the FLC-curve). All measurements (n = 3) were conducted at the growth temperature of 2 °C.

## 2.4 Pigments

Samples for the determination of pigment concentration were filtered onto GF/F filters and immediately frozen at -80 °C until further analysis. Pigments samples were homogenized and extracted in 90% acetone for 24h at 4 °C in the dark. After centrifugation (5 min, 4 °C, 13000 rpm) and filtration through a 0.45 µm pore size nylon syringe filter (Nalgene®, Nalge Nunc International, Rochester, NY, USA), concentrations of chlorophyll $a$ (Chl $a$) and $c_2$ (Chl $c_2$), and alloxanthin (Allo) were determined by reversed phase High Performance Liquid Chromatography (HPLC). The analysis was performed on a LaChromElite® system equipped with a chilled autosampler L-2200 and a DAD detector L-2450 (VWR-Hitachi International GmbH, Darmstadt, Germany). A Spherisorb® ODS-2 column (25 cm x 4.6 mm, 5 $\mu$m particle size; Waters, Milford, MA, USA) with a LiChropher® 100-RP-18 guard cartridge was used for the separation of pigments, applying a gradient according to Wright et al. (1991). Peaks were detected at 440 nm and identified as well as quantified by co-chromatography with standards for Allo, Chl $a$ and c2 (DHI Lab Products, Hørsholm, Denmark) using the software EZChrom Elite ver. 3.1.3. (Agilent Technologies, Santa Clara, CA, USA). Pigment contents were normalized to filtered volume and cell densities to yield cellular quotas.

## 2.5 Statistics

Combined effects of the two $pCO_2$ (ambient and OA) and light (LL, ML, and HL) treatments on all experimental parameters were statistically analyzed using two-way analyses of variance (ANOVA) with Bonferroni's multiple comparison post tests. To test for significant differences between light treatments of the OA-grown cells of *G. cryophila* cells one-way ANOVAs with additional Bonferroni's multiple comparison post tests were applied. All statistical analyses were performed using the program GraphPad Prism (Version 5.00 for Windows, Graph Pad Software, San Diego California, USA) and the significance testing was done at the $p < 0.05$ level.

## 3 Results

### 3.1 Seawater carbonate chemistry

The two target $pCO_2$ levels of 400 and 1000 µatm were successfully achieved for abiotic control (abiotic, bubbled seawater without cells) and culture bottles (biotic, 1-*way ANOVA*: $p < 0.0001$, Table 1). As the $pCO_2$ of the abiotic control and culture bottles of the same $pCO_2$ treatment were similar, this indicates that final cell numbers of *G. cryophila* did not alter the $pCO_2$ of the culture bottles relative to the culture medium. Similar trends as for the $pCO_2$ were also apparent for the measured pH values, which yielded 8.13 ± 0.07 and 7.82 ± 0.06 in the culture bottles of the ambient and OA treatments, respectively (Table 1). While DIC concentrations were significantly enhanced for

OA relative to the ambient $pCO_2$ treatments (*1-way ANOVA*: $p < 0.0001$), they also significantly differed between abiotic control (abiotic, bubbled seawater without cells) and culture bottles (biotic, Table 1).

**3.2 Growth, elemental stoichiometry and composition**

Growth rates were significantly affected by light (*2-way ANOVA*: $p < 0.01$), but not by OA or the interaction of both factors (Fig. 1A). In response to increasing irradiance, growth rates of cells grown under ambient $pCO_2$ remained unchanged between LL and ML, but were negatively influenced by HL as they were unable to grow. Under OA, however, growth rates significantly increased between LL and ML by 89% (posthoc: $p < 0.05$) and between ML and HL by 32% (posthoc: $p < 0.05$), respectively. Irrespective of changes in irradiance, $pCO_2$ or their combination, cellular POC contents did not change (Fig. 1B). Daily POC production rates, were, however, significantly altered by increasing irradiance (*2-way ANOVA*: $p < 0.01$), but not by OA or the interaction of both factors (Fig. 1C). While increasing light intensity did not affect POC production rates of the ambient $pCO_2$ treatments, there was a significant OA-dependent enhancement by 69% between LL and ML (posthoc: $p < 0.01$) and by 39% between ML and HL, respectively (posthoc: $p < 0.05$). Molar C:N ratios were only significantly influenced by the interaction of both factors together (*2-way ANOVA*: $p < 0.01$; Fig. 1D). From LL to ML C:N ratios did not change for both $pCO_2$ treatments whereas from ML to HL the ratio declined by 12% for the OA treatment (*1-way ANOVA*: $p < 0.05$). In response to increasing $pCO_2$, C:N decreased by 10% when grown under LL (posthoc: $p < 0.05$), but remained unaltered at ML.

**3.3 Pigments**

For all $pCO_2$ treatments, cellular concentrations of the measured pigments (Allo, Chl *a* and $c_2$) showed a strong and significant decline between LL and ML (*2-way ANOVA*: $p < 0.0001$, Table 2). Between ML and HL, however, different effects were seen, with a significant enhancement for Chl *a* (posthoc: $p < 0.01$) and Allo (posthoc: $p < 0.01$) and no effect for Chl $c_2$. Increasing $pCO_2$ had generally no effect on cellular pigment quotas (Allo, Chl *a* and $c_2$) except for the Allo quotas of the LL treatments, which displayed a significant OA-dependent decline by 26% (posthoc: $p < 0.01$). A significant interaction between light and $CO_2$ was only found for Allo quotas (*2-way ANOVA*: $p < 0.05$).

**3.4 Chlorophyll *a* fluorescence**

The dark-adapted maximum quantum yield of PSII ($F_v/F_m$) was strongly influenced by irradiance (*2-way ANOVA*: $p < 0.0001$) and $CO_2$ (*2-way ANOVA*: $p = 0.0012$) and their interaction (*2-way ANOVA*: $p < 0.05$; Fig. 2A). With increasing irradiance $F_v/F_m$ generally declined whereas OA increased it at LL (17%, posthoc: $p < 0.01$) or did not change it at ML. Noticeably, the interaction of HL and OA resulted in the lowest $F_v/F_m$ value. Comparing the $F_v/F_m$ measured before and after the FLC-curve, the $F_v/F_m$ recovery was calculated and given as % of the initial $F_v/F_m$. Neither high $pCO_2$ nor the interaction of light and $CO_2$ affected $F_v/F_m$ recovery whereas the increase in irradiance had a significant effect (*2-way ANOVA*: $p < 0.01$, Fig. 2B), being increased by 11% between LL and ML under ambient $pCO_2$ (posthoc: $p < 0.05$).

The increase of $CO_2$ or light alone had no effect on cellular concentrations of functional photosystem II reaction centers ([RCII]) while the interaction of both factors strongly altered [RCII] (*2-way ANOVA*: $p < 0.0001$; Fig. 3A). From LL to ML [RCII] decreased under ambient $pCO_2$ (39%, posthoc: $p < 0.001$) while the combination of ML with OA synergistically increased it (44%, posthoc: $p < 0.01$, Fig. 3). [RCII] was reduced by OA at LL (37%

posthoc: $p < 0.01$) whereas the combined effect of OA and ML led to an increase (49%, posthoc: $p < 0.01$). While $CO_2$ and the interaction of $CO_2$ and light together did not change the energy transfer between PSII units (i.e. connectivity, $P$), only the increase in irradiance had a significant effect (*2-way ANOVA*: $p < 0.05$), reducing $P$ by 22% between LL and ML under OA (posthoc: $p < 0.05$, Figure 3B). While the increase in $CO_2$ or light did not alter the functional absorption cross-sections of PSII ($\sigma_{PSII}$), the interaction of both factors, however, had an effect (*2-way ANOVA*: $p < 0.05$; Figure 3C). $\sigma_{PSII}$ values were similar under LL and ML at ambient $pCO_2$. The interaction of OA and ML, however, lowered them (posthoc: $p < 0.05$, Table 3). On the other hand, when grown under OA $\sigma_{PSII}$ was larger under HL than under ML (*1-way ANOVA*: $p < 0.01$). Re-oxidation times of the primary electron acceptor $Q_a$ ($\tau_{Qa}$) significantly changed with increasing irradiance (*2-way ANOVA*: $p < 0.05$), but not by high $CO_2$ or the interaction of both factors together (Figure 3D). $\tau_{Qa}$ of OA-acclimated cells was much shorter at HL than at ML (*1-way ANOVA*: $p < 0.05$).

Absolute ETRs differed in amplitude and shape in response to the applied changes in irradiance and $pCO_2$ (Fig. 4A-C). Both maximum absolute electron transport rates ($ETR_{max}$) and minimum saturating irradiances ($I_K$) followed the same trend and were significantly changed by $CO_2$ (*2-way ANOVA*: $p < 0.05$), but not by light or the interaction of both factors (Table 3). OA significantly enhanced both parameters under LL ($ETR_{max}$: posthoc: $p < 0.05$, $I_K$: posthoc: $p < 0.05$), but not under ML. The maximum light utilization efficiency ($\alpha$) was significantly affected by light (*2-way ANOVA*: $p < 0.01$) and the interaction of $CO_2$ and light (*2-way ANOVA*: $p < 0.01$), but not by $CO_2$ alone (Table 3). $\alpha$ significantly increased from LL to ML at ambient $pCO_2$ (53%, posthoc: $p < 0.01$) while such effect was absent under ML and OA. Between ML and HL, $\alpha$ did not differ when grown under OA.

Nonphotochemical quenching (NPQ) generally went up with increasing actinic irradiance during the FLC (Fig. 4D-F). Compared with the LL treatments, NPQ values of the ML and HL treatments were as twice as high. There were no differences in the NPQ pattern between ML and HL treatments observed. Much higher NPQ values were determined in the ambient $pCO_2$ relative to the OA treatment under LL while such $pCO_2$ effect was absent under ML.

## 4 Discussion

Ecophysiological studies on Antarctic cryptophytes to assess whether climatic changes such as ocean acidification and enhanced stratification affect their growth in Antarctic coastal waters in the future are lacking so far. This study can show that the Antarctic cryptophyte *G. cryophila* may be a potential winner of such climatic conditions as it reached highest rates of growth and particulate organic carbon production when grown under HL and OA.

### 4.1 *Geminigera cryophila* is sensitive to increasing irradiance under ambient $pCO_2$

The cryptophyte *G. cryophila* was well adapted to grow under LL and ML at ambient $pCO_2$, yielding similar growth, POC quotas and production rates as well as C:N ratios (Fig. 1). In line with this, the exposure of the cryptophyte *Rhodomonas salina* to 30 up to 150 $\mu$mol photons m$^{-2}$ s$^{-1}$ did not lead to any changes in growth rate at 5 °C (Hammer et al., 2002). Even though growth and biomass remained unchanged in *G. cryophila* between LL and ML, acclimation to the even higher light intensity of 500 $\mu$mol photons m$^{-2}$ s$^{-1}$ was indicated by the reduction in $F_v/F_m$ and [RCII] (Figs. 2A, 3A). Such decline in the number of photosystems is a typical photoacclimation response of most microalgae to increasing light and is usually accompanied by a decrease in cellular concentrations of light harvesting pigments (MacIntyre et al., 2002), as seen here for cellular Chl $a$ and c$_2$ quotas (Table 2). Even though

most studies on temperate cryptophytes report a photoprotective function of Allo, with higher amounts of this carotenoid toward high irradiance (Funk et al., 2011; Laviale and Neveux, 2011), the reduction in cellular Allo quotas from LL to ML in our tested species (Table 2) rather suggests its role in light absorption. Similarly, cellular Allo contents also declined between 40 and 100 $\mu$mol photons m$^{-2}$ s$^{-1}$ in the temperate cryptophyte *Rhodomonas marina* (Henriksen et al., 2002). In this study, various photophysiological parameters ($P$, $\sigma_{PSII}$, $\tau_{Qa}$, Fig. 3) did not change between LL and ML in *G. cryophila* while other photoacclimation processes such as higher ETR$_{max}$, $I_k$ and $\alpha$ took place (Table 3). Such light-dependent apparent higher electron flow was accompanied by similar high POC quotas and production rates between LL and ML (Fig. 1B, C) and suggests saturation of the Calvin cycle and therewith the requirement for alternative electron cycling to dissipate the excessive light energy. In the temperate *R. salina*, the onset of NPQ was induced after saturation of the Calvin cycle and found to be located in the chlorophyll a/c proteins and not in the phycobiliproteins (Kana et al., 2012). In fact, NPQ was strongly enhanced in *G. cryophila* between LL and ML at ambient pCO$_2$ (Fig. 4). As ML- relative to LL-acclimated cells also exhibited a higher potential of $F_v/F_m$ recovery after the FLC curve (Fig. 2B), it appears that all these adjustments allowed a reduction of the excitation pressure on the photosynthetic apparatus and protected *G. cryophila* well against short-term high light exposure. Unexpectedly, *G. cryophila* was, however, unable to grow at ~~the highest light intensity of~~ 500 $\mu$mol photons m$^{-2}$ s$^{-1}$ under ambient pCO$_2$, pointing towards its vulnerability to cope with HL under present day pCO$_2$. On the other hand, long-term field observations have shown that cryptophytes mainly occur under stratified conditions along the WAP (e.g. Moline and Prézelin, 1996; Moline et al., 2004; Mendes et al., 2013). A connection of this group with high illuminated conditions was first suggested by Mendes et al. (2017), but lacks information which cryptophyte species were present and their photosynthetic responses. The reason for this difference could be related to species- or strain-specific differences. On the basis of our results, the here tested *G. cryophila* strain was indeed able to cope well with medium, but not high irradiances. More tests with other cryptophytes are certainly required for being able to better evaluate cryptophytes' abilities to cope with high irradiances.

**4.2 OA alters the physiological response of *G. cryophila* to high irradiance**

In line with previous studies on Antarctic diatoms and *P. antarctica* (Hoogstraten et al., 2012; Heiden et al., 2016; Trimborn et al., 2017b), OA in conjunction with low irradiance did not alter growth, cellular contents or production rates of POC in *G. cryophila* (Fig. 1A-C). There was, however, an OA-dependent decline in C:N under LL (Fig. 1D), resulting from a significant enhancement of PON quotas between ambient and high pCO$_2$ (23.1 ± 1.4 and 29.0 ± 1.6 pg N cell$^{-1}$, respectively). Hence, *G. cryophila* cells most probably benefitted from lower energy investments to acquire inorganic carbon under high pCO$_2$. Whereas the two temperate cryptophytes *Rhodomonas* sp. and *Chroomonas* sp. (Burns and Beardall, 1987; Camiro-Vargas et al., 2005), similar to other Antarctic phytoplankton taxa (Trimborn et al., 2013), were able to actively take up CO$_2$ and HCO$_3^-$, the operation of a carbon concentrating mechanism in *G. cryophila* has not been tested so far. As typically observed for temperate phytoplankton (Hopkinson et al, 2011; McCarthy et al., 2012; Yang and Gao, 2012), the higher pCO$_2$ was not used to fix more POC per cell by *G. cryophila* at LL (Fig. 1B, C), but instead fuelled protein build-up through conversion of carbohydrate skeletons to proteins. N assimilation is energetically costly due to the reduction steps involved (Sanz-Luque et al., 2015), therefore the finding of elevated PON buildup under OA and LL is surprising. In line with this, ETR$_{max}$ per PSII increased by 121% from ambient to high pCO$_2$ (Table 3), likely used to reduce nitrite to ammonium. Calculating overall maximum ETRs per cell (cETR$_{max}$ = ETR$_{max}$ x [RCII], given in amol cell$^{-1}$ s$^{-1}$), there

was also an OA-dependent increase in $cETR_{max}$ by 41% (mean value of 76 and 107 amol cell$^{-1}$ s$^{-1}$ under ambient and high $pCO_2$, respectively), but this increase was comparably lower relative to $ETR_{max}$ per PSII. The reason for this comes from the strong reduction of [RCII] between ambient and high $pCO_2$ at LL (Fig. 3). Next to the positive OA effect on N metabolism, also the photochemical efficiency ($F_v/F_m$) of *G. cryophila* was significantly increased by 17% by OA under LL (Fig. 2A). Unexpectedly, this effect was not the result of reduced cellular quotas of the light harvesting pigments (Chl *a* and c$_2$), as they remained the same under these conditions. Instead a significant OA-dependent decrease by 26% in the carotenoid Allo was found (Table 2), which could explain the positive OA effect on $F_v/F_m$. This is in line with the OA-dependent reduction in NPQ observed in LL-acclimated *G. cryophila* cells (Fig. 4), pointing towards a reduced need to dissipate excess light energy following short-term high light exposure. Overall, OA in conjunction with LL was beneficial for *G. cryophila*, with positive effects in particular on N metabolism.

The beneficial OA effect on N assimilation under LL, however, vanished at ML (Fig. 1D, 23.2 ± 3.2 and 23.2 ± 2.3 pg N cell$^{-1}$, respectively), probably as a result from the higher N metabolism cost to maintain photosynthesis under these conditions (Li et al., 2015). Based on our results, the physiology of the cryptophyte *G. cryophila* remained more or less unchanged between ambient and high $pCO_2$ at ML (Figs.1, 2, 4, Tables 2 and 3). At the highest irradiance (HL), *G. cryophila* could not grow under ambient $pCO_2$, but surprisingly grew well under the same light intensity in conjunction with OA, displaying highest growth rates compared to all other treatments (Fig. 1A). Similarly, this species also showed highest production rates of POC and PON (Fig. 1). Compared with the tested Antarctic diatoms and *P. antarctica* so far, which exhibited either negative or neutral effects in response to OA and high irradiance on growth and/or photosynthesis (Feng et al., 2010; Heiden et al. 2016, Trimborn et al. 2017a,b; Heiden et al. 2019; Koch et al. 2019), growth and photosynthesis of the cryptophyte benefitted synergistically from OA and HL. Looking at the significantly higher amounts of Chl *a* and Allo per cell and its larger $\sigma_{PSII}$ between ML and HL under OA (Table 2, Figure 3C), this species even reinforced its capacity to absorb light. Faster electron drainage into downstream processes was evident by the shorter Qa re-oxidation time between ML and HL under OA (Figure 3D), supporting that this species indeed managed well to cope with these conditions.

**4.3 Implications for the ecology of *G. cryophila* in future coastal Antarctic waters**

Along the coast of the Western Antarctic Peninsula, diatoms, prymnesiophytes and cryptophytes represent the main phytoplankton groups, which form prominent blooms and therefore strongly contribute to carbon biomass build-up (Garibotti et al., 2005; Trimborn et al., 2015). Occurrence of cryptophytes in this region was associated with low salinity and warm stratified surface waters (Moline and Prézelin, 1996; Moline et al., 2004; Mendes et al., 2013; Mendes et al., 2017). Only recently, it was suggested that a high tolerance of cryptophytes to withstand high irradiances could potentially explain their occurrence in well illuminated surface waters by Mendes et al. (2017). Our results from a short-term $CO_2$-light experiment point towards a high ability of *G. cryophila* to acclimate to such conditions and to cope well with medium, but not high irradiances, whether this applies for other Antarctic cryptophyte species as well needs further testing. Also it remains unclear whether similar responses would be found when exposed on a longer term. In general, with respect to the projected climatic changes little is known about the potential $CO_2$ sensitivity of cryptophytes. Previous studies mainly assessed the response to OA on cryptophytes at the community level and showed no discernible effects on their abundance (Domingues et al., 2014; Sommer et al., 2015; Young et al., 2015; Schulz et al., 2017; Donahue et al., 2019). This study is the first at the species level to

show that the combination of OA and high irradiance promoted growth and biomass production in the Antarctic cryptophyte *G. cryophila*. In fact, while HL conditions inhibited growth of this species under ambient $pCO_2$, the combination of OA and HL, on the other hand, enabled it to grow and to cope even better with the applied environmental conditions, reaching highest growth and POC production rates (Fig. 1). This was also accompanied with a high photophysiological capacity of this species when exposed on the short-term to increasing irradiances. The beneficial effect of OA and HL for *G. cryophila* is opposed to previous observations, where growth and/or photosynthesis was inhibited in several diatoms, but no effect for the prymnesiophytes *P. antarctica* (Feng et al., 2010; Trimborn et al., 2017a,b; Beszteri et al., 2018; Heiden et al., 2018; Koch et al., 2018, Heiden et al., 2019). Hence, *G. cryophila* could be a potential winner of climate change, with higher abundances and increased contribution to the productivity of future stratified, acidified and well illuminated coastal Antarctic waters. This study further confirms previous results (Moline et al., 2004; Ducklow et al., 2007; Montes-Hugo et al., 2009; Mendes et al., 2017), which point towards a stronger importance of flagellates in the future. A functional shift away from efficient carbon sinkers such as diatoms to less efficient carbon vectors such as flagellates including cryptophytes and prymnesiophytes could, however, diminish the strength of the biological carbon pump of future Antarctic coastal waters.

**Author contributions**

ScTr designed the study. PK conducted the experiment. ScTr, SiTh and PK analysed the data. ScTr prepared the paper with contributions from SiTh, PK and KB.

**Acknowledgements**

Helen Soares de Souza, Jasmin P. Heiden, Tina Brenneis and Britta Meyer are thanked for the support in the laboratory. ScTr and PK were funded by the Helmholtz association (Young Investigator Group *EcoTrace*, VH-NG-901).

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

Table 1 Partial pressures of $CO_2$ ($pCO_2$) for the ambient and ocean acidification (OA) treatments were calculated from measured pH, concentrations of dissolved inorganic carbon (DIC), silicate and phosphate, temperature, and salinity using the CO2Sys program (Pierrot et al. 2006). For all parameters, values are given for the abiotic control bottles (abiotic, bubbled seawater without cells) and the culture bottles at the end of the experiment (biotic). Different letters indicate significant differences between treatments ($p < 0.05$). Different letters indicate significant differences between treatments ($p < 0.05$). Values represent the means $\pm$ SD (n = 3).

| Target $pCO_2$ (µatm) | $pCO_2$ (µatm) | | pH (NBS) | | DIC (µmol kg$^{-1}$) | |
|---|---|---|---|---|---|---|
| | abiotic | biotic | abiotic | biotic | abiotic | biotic |
| Ambient $pCO_2$, 400 | $372 \pm 18^a$ | $398 \pm 66^a$ | $8.14 \pm 0.02^a$ | $8.13 \pm 0.07^a$ | $2024 \pm 7^a$ | $2062 \pm 12^b$ |
| OA, 1000 | $986 \pm 42^b$ | $865 \pm 120^b$ | $7.75 \pm 0.02^b$ | $7.82 \pm 0.06^b$ | $2160 \pm 7^c$ | $2195 \pm 16^d$ |

Table 2 Cellular concentrations of chlorophyll $a$ and $c_2$ (Chl $a$ and $c_2$) as well as alloxanthin (Allo) were determined for *Geminigera cryophila* acclimated to ambient or high $CO_2$ conditions combined with low (LL), medium (ML) or high light (HL). *G. cryophila* did not grow under ambient $pCO_2$ and HL as indicated by ng. Photosynthetic parameters were derived from at least three independent measurements. Different letters indicate significant differences between treatments ($p < 0.05$). Values represent the means $\pm$ SD.

| Treatment | Chl $a$ (fg cell$^{-1}$) | Chl $c_2$ (fg cell$^{-1}$) | Allo (fg cell$^{-1}$) |
|---|---|---|---|
| Ambient pCO$_2$ LL | 2358 $\pm$ 277[a] | 172 $\pm$ 33[a] | 81 $\pm$ 3[a] |
| OA LL | 2300 $\pm$ 25[a] | 131 $\pm$ 1[a] | 60 $\pm$ 2[b] |
| Ambient pCO$_2$ ML | 871 $\pm$ 225[b] | 75 $\pm$ 24[b] | 30 $\pm$ 8[c] |
| OA ML | 662 $\pm$ 22[b] | 48 $\pm$ 15[b] | 27 $\pm$ 3[c] |
| Ambient pCO$_2$ HL | ng | ng | ng |
| OA HL | 772 $\pm$ 85[c] | 41 $\pm$ 1[b] | 36 $\pm$ 1[d] |

Table 3 Maximum absolute electron transport rates ($ETR_{max}$), minimum saturating irradiances ($I_K$) and maximum light utilization efficiencies (α) were determined for *Geminigera cryophila* acclimated to ambient or high $CO_2$ conditions combined with low (LL), medium (ML) or high light (HL). *G. cryophila* did not grow under ambient $pCO_2$ and HL as indicated by ng. Photosynthetic parameters were derived from at least three independent measurements. Different letters indicate significant differences between treatments ($p < 0.05$). Values represent the means ± SD.

| Treatment | $ETR_{max}$ (e$^-$ PS$^{-1}$ s$^{-1}$) | $I_k$ ($\mu$mol photons m$^{-2}$ s$^-$) | α (rel. unit) |
|---|---|---|---|
| Ambient $pCO_2$ LL | 126 ± 28[a] | 72 ± 9[a] | 1.75 ± 0.17[a] |
| OA LL | 279 ± 52[b] | 142 ± 34[b] | 1.98 ± 0.11[a] |
| Ambient $pCO_2$ ML | 265 ± 77[b] | 102 ± 38[b] | 2.67 ± 0.26[b] |
| OA ML | 278 ± 70[b] | 138 ± 48[b] | 2.06 ± 0.24[a] |
| Ambient $pCO_2$ HL | ng | ng | ng |
| OA HL | 379 ± 38[b] | 156 ± 40[b] | 2.50 ± 0.37[a] |

**Figure legend**

Figure 1 Growth rate (A), cellular content (B) and production rate (C) of particulate organic carbon (POC) and the molar ratio of carbon to nitrogen (C:N, D) for *Geminigera cryophila* acclimated to ambient (black bars) or high $CO_2$ (grey bars) conditions combined with low (LL), medium (ML) or high light (HL). *G. cryophila* did not grow under ambient $pCO_2$ and HL as indicated by ng. Values represent the means ± SD (n = 3). Different letters indicate significant differences between treatments ($p < 0.05$).

Figure 2 The dark-adapted maximum PSII quantum yield $F_v/F_m$ (A) and the yield recovery after short-term light stress (% of initial) (B) for *Geminigera cryophila* acclimated to ambient (black bars) or high $CO_2$ (grey bars) conditions combined with low (LL), medium (ML) or high light (HL). *G. cryophila* did not grow under ambient $pCO_2$ and HL as indicated by ng. Values represent the means ± SD (n = 3). Different letters indicate significant differences between treatments ($p < 0.05$).

Figure 3 Cellular concentrations of functional photosystem II reaction centers [RCII] (A), energy transfers between PSII units (i.e. connectivity, *p*) (B), functional absorption cross sections of PSII photochemistry ($\sigma_{PSII}$) (C) and re-oxidation times of the primary electron acceptor $Q_a$ ($\tau_{Qa}$) (D) were determined for *Geminigera cryophila* acclimated to ambient or high $CO_2$ conditions combined with low (LL), medium (ML) or high light (HL). *G. cryophila* did not grow under ambient $pCO_2$ and HL as indicated by ng. Different letters indicate significant differences between treatments ($p < 0.05$).

Figure 4 Absolute electron transport rates (ETR, A-C) and nonphotochemical quenching (NPQ, D-F) were measured in response to increasing irradiance in *Geminigera cryophila* acclimated to ambient (black circles) or high $CO_2$ (white circles) conditions combined with A) low (LL), B) medium (ML) or C) high light (HL). *G. cryophila* did not grow under ambient $pCO_2$ and HL. ETRs were obtained in three individual measurements. Values represent the means ± SD (n = 3).

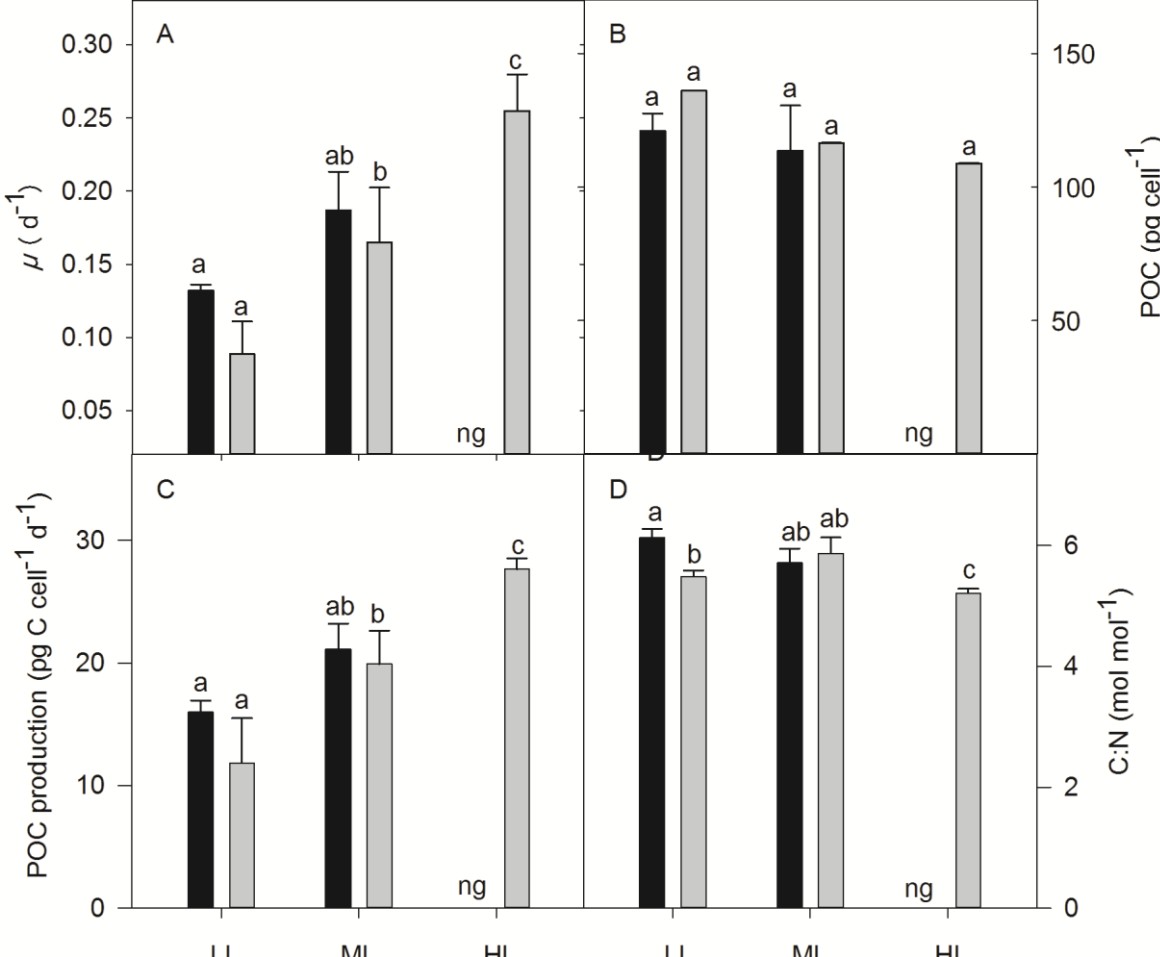

Figure 1

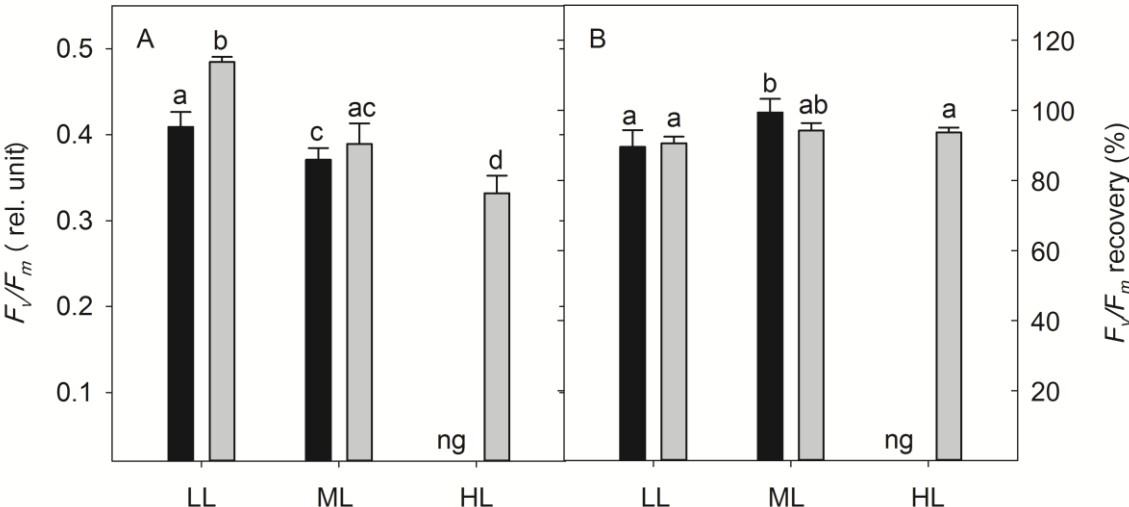

Figure 2

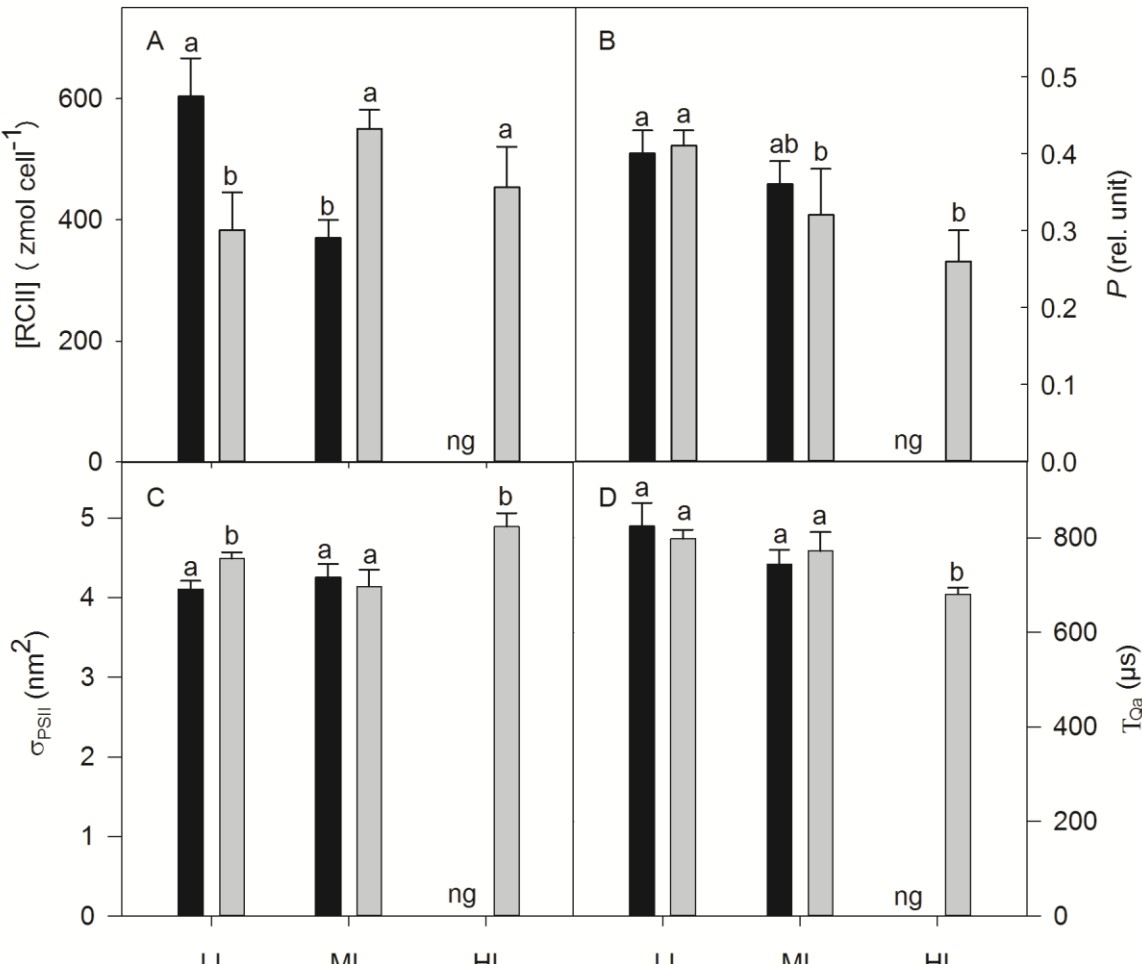

Figure 3

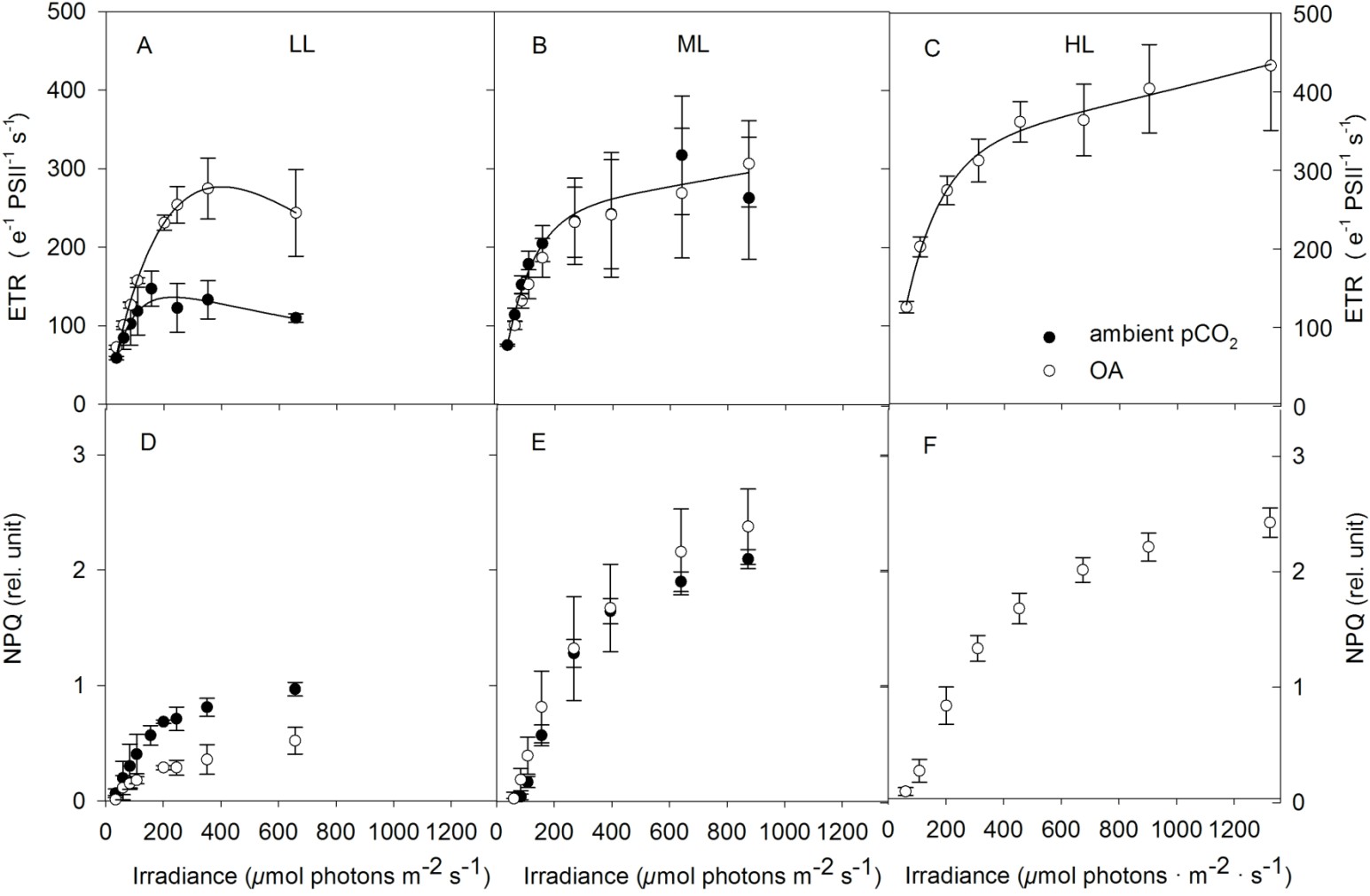

Figure 4