# Peer review of "Ocean acidification and high irradiance stimulate the photophysiological fitness, growth and carbon production of the Antarctic cryptophyte *Geminigera cryophila"

_Biogeosciences, 2019_

## Referee Comment (RC2)

Marco J. Cabrerizo
2019-04-16
10.5194/bg-2019-97-RC2
journal article
en umol L-1) and nitrate (final concentration 6.25 umol L-1) (N:P ratio of 16:1, Red-field 1963), as well as, trace metals and vitamins according to F/2 medium (Ref). Cells were exposed under a 16h L: 8h D cycle and three constant light intensities (20, 200 and 500 umol m-2 s-1) (LL, ML and HL, respectively)".

Once you have already described the pre-acclimation phase, you can describe your experimental setup, and finally, all the details related with the CO2 enrichment. For example, "To assess the interactive effect of light and OA on the photophysiology, metabolism and growth of G. cryophila, a 3x2 full factorial matrix (in triplicate) was implemented with: (a) three light levels (LL, ML and HL) and two pCO2 treatments (ambient and OA)"...

Line 15: Cells in exponential or mid-exponential phase. Please clarify.

Statistics: Were ANOVA assumptions checked?. Was the interaction light and OA significant in all response variables? Please clarify. Over the results section, I cannot see if Light×OA was or not significant.

Results: I propose authors to change the ordering of the results. First, they should show PSII variables, then pigments, metabolism and stoichiometry and finally growth. The rationale behind my proposal is related with the fact that changes at PSII level occurs in temporal scales ranging mseg to minutes, whereas those related with the other variables need more time.

Line 30: In my humble opinion, I think that authors should consider the effect of light under ambient pCO2. Authors state that "growth rates remained unchanged in cells grown under ambient pCO2", however, according to the figure 1, increasing light had a negative effect on cells. In fact, they did not grow.

Line 38: Change significantly influenced by negatively influenced Chl a fluorescence (Lines 12-13): I would rewrite this paragraph in a more direct style. For example, it could be rewritten as follow: "Regarding to the individual effects of both factors on

PSIImax, it was found that whereas light conditions decreased it (particularly at HL), OA increased (ca. 17% at LL) or did not affect it (i.e. at ML). Noticeably, lowest values were found under the LightxOA interaction (ca. 0.35) (Fig. 2A)".

Chl a fluorescence (Lines 20-21): This paragraph is a little bit confusing. To avoid any misunderstanding I would rephrase this section as follow: "Fv/Fm recovery was not influenced neither by light nor pCO2 treatment, excepting under ambient pCO2 and ML treatments in which the quantum yield's recovery was maxima (Fig. 2A)".

Chl a fluorescence (Lines 22-25): Similarly than mentioned in the previous paragraphs, such idea could be simplified as: "For [RCII], increasing light conditions reduced a 39% the functional RC; however, OA did not show a clear response pattern, as it both decreased ($\sim$XX% at LL) as increased ($\sim$44% at ML) the functional RC (Fig. 3A)".

Chl a fluorescence (Lines 26-34): In my humble opinion, in this paragraph is difficult extracting the main finding, and what is the effect of light, OA and their interaction. Also, I sincerely think that authors should present this dataset at the same way than for figure 1-3, in a bar chart. If I see the table 3 presented, I would summarize the findings as follow: "In relation to the individual effects of light and OA on the PSII photophysiology it was found that increasing light conditions exerted a mostly stimulatory effect, excepting on connectivity (i.e. P) where it was inhibitory. OA had a significant positive effect at LL on $\sigma$PSII, ETRmax and Ik, but significantly inhibitory on $\alpha$ at ML. Light and OA, as a single factors, did not exert any significant effect on P and $\tau$Qa. At the same way than mentioned above, the LightxOA interaction had a contrasting effect on the PSII-photosphysiology, being stimulatory on $\sigma$PSII, ETRmax and Ik, and inhibitory on P and $\tau$Qa. The interactive effect of both factors did not alter $\alpha$".

Chl a fluorescence (Lines 37): I suggest that figure 4 could be moved to supplementary information, as the core information related with this section/variable is contained in Table 3. Thus, and similarly than mentioned above to authors, Table 3 should be presented as figure instead table. Regarding to the Fig.4, authors can highlight into the text, the stimulatory effect of OA on ETR under LL, and the absence of effect by OA under ML. Also, that such stimulatory effect was coupled with significant? lower NPQ values (i.e. LL) or with not significant differences between amb and OA treatments (i.e. ML). Maxima ETR and NPQ values were found under HL.

Regarding to the figure 5, I suggest that it could be also moved to supplementary information together with figure 4, and merge both figures into a single one but with 6 panels.

Finally, despite the main goal of this study was assess the interactive effect of HL and OA, authors did not explicitly quantified such impacts on the response variables assessed. Considering that only the OA effect could be calculated, as cells did not grow under HL, I suggest authors calculating the size effect in percentage (or the log response ratio) at least for the HLxOA interaction and for all variables, and show them together in a final figure. Using this approach could strengthen the message of your work, and easily showing to a potential reader if the HLxOA interaction was synergistic/antagonistic (or stimulatory/inhibitory), and on what variable(s).

Discussion: What is the main finding of your study?. It would be nice highlighting to a potential reader in a sentence or a paragraph what is the gap that this study fills (or contributes).

Lines 28-29: Based on the fact that significant higher ETRmax, Ik and alpha values were not coupled with higher POC, and ultimately growth, could it be plausible that cells were less efficient than under LL? Or more damaged?. Thus, this lower photosynthetic efficiency (or higher damage) would be consistent with a 2-fold increase in the NPQ, and ultimately, could support the higher Fv/Fm recovery (%).

Line 34: I think that "high light" should be modified to avoid any misunderstanding with the HL treatment.

Line 40: As the inability for growing of the target specie is a surprising result, and previous results have found that such phytoplankton group grow under HL and stratified conditions, I suggest that authors discuss the potential mechanism(s) or reason(s) that impeded that cell grown.

Lines 46 and 69: Statistical information can be omitted in discussion section

Implications: On one hand, it is true that the interaction HLxOA stimulated all assessed processes in G. cryophila, however, such responses were based on a short-term scale. It could be plausible that such beneficial effects would be accentuated at long-term scales by an adaptation of the populations, or by contrast, reduced. Authors could briefly discuss such issue in the context of their study.

On the other hand, in my humble opinion, I think that this idea presented at the end of the manuscript should be also highlighted into the Introduction section, as it represents the core about why assess the interactive effects of both global-change factors on Antarctic phytoplankton communities. Currently, we do not know how communities will respond to such changes, if such changes will lead to a diatoms or flagellates-dominated community, and as a consequence, if it will boost or reduce the C-sink capacity of this area.

Minor comments:

Tables 1-3: Please, include what lowercase letters mean. Are they representing posthoc comparisons?. Data represent means and SD?. And OA?. Introduction (line 54): Change almost nothing by little is known

Please also note the supplement to this comment:
https://www.biogeosciences-discuss.net/bg-2019-97/bg-2019-97-RC2-supplement.pdf

**Supplement:**

Trimborn et al. evaluate in a full-factorial experimental design how the
interaction between ocean acidification and high light alter the photophysiology,
stoichiometry, production and growth of a model Antarctic Cryptophyte at short-term
scales.  The topic assessed fits with the *Biogeoscience's* scope, and is novel as currently
information related with the interactive effect of both factors on Antarctic
phytoplankton is still scarce.  The experiment is well-designed and performed, and the
results are appropriate to being published in the Journal.

Below, authors will find a point by point revision with the main issues found
over the manuscript, and suggestions which I hope that they find useful.

Respectfully submitted,

Marco J. Cabrerizo

**Title:** I suggest including in the title two variables also quantified by authors:
production and photophysiology of the species.  On potential title could be: "*Ocean*
*acidification and high irradiance stimulate the photophysiology, growth and production*
*in the Antarctic cryptophyte Geminigera cryophila*".

**Introduction:** What are the individual effects of HL and OA on phytoplankton?.
I suggest that authors add some general information about the individual effects of light
and OA on primary producers, and then they focus such impacts on Antarctic
phytoplankton and crytophytes.  Through a general view, a potential reader can identify
the gaps of knowledge related with the quantification of such impacts on phytoplankton,
and the scarcity of experimental studies testing their interaction on this key group.

**Lines 27-29:** If it was found a contrasted response pattern in southern WAP, it
means that there were both a positive as a negative effect. In such case, it partially
agrees with those results reported in the northern part. Please rewrite.

**Lines 29-30:** Break this sentence to separate both ideas.

**Line 45:** Reference?

**Line 69:** I think that it would be nice that authors include a hypothesis work
about what do they expect based on the previous information known?.

**Material and Methods:**

**Culture conditions:** This subsection is confusing at the present state. I propose
you modifying it as follow: "*Before to being used in experimentation (two weeks),*
*triplicate semi-continuous cultures of the Antarctic cryoptophyte G. cryophyla (CCMP*
*2564) were grown (and maintained in mid-exponential growth) at 2ºC in sterile -filtered*
*(0.2 um) Antarctic seawater (salinity 30.03) enriched with phosphate (final*
*concentration 100 umol L-1) and nitrate (final concentration 6.25 umol L-1) (N:P ratio*
*of 16:1, Redfield 1963), as well as, trace metals and vitamins according to F/2 medium*

*(Ref).  Cells were exposed under a 16h L: 8h D cycle and three constant light intensities*
*(20, 200 and 500 umol m-2 s-1) (LL, ML and HL, respectively)".*

Once you have already described the pre-acclimation phase, you can describe
your experimental setup, and finally, all the details related with the CO2 enrichment.
For example, "*To assess the interactive effect of light and OA on the photophysiology,*
*metabolism and growth of G. cryophila, a 3x2 full factorial matrix (in triplicate) was*
*implemented with: (a) three light levels (LL, ML and HL) and two pCO2 treatments*
*(ambient and OA)*"…

**Line 15:** Cells in exponential or mid-exponential phase. Please clarify.

**Statistics:** Were ANOVA assumptions checked?. Was the interaction light and
OA significant in all response variables? Please clarify. Over the results section, I
cannot see if Light×OA was or not significant.

**Results:** I propose authors to change the ordering of the results. First, they
should show PSII variables, then pigments, metabolism and stoichiometry and finally
growth.  The rationale behind my proposal is related with the fact that changes at PSII
level occurs in temporal scales ranging mseg to minutes, whereas those related with the
other variables need more time.

**Line 30:** In my humble opinion, I think that authors should consider the effect of
light under ambient pCO2. Authors state that "*growth rates remained unchanged in*
*cells grown under ambient pCO2*", however, according to the figure 1, increasing light
had a negative effect on cells. In fact, they did not grow.

**Line 38:** Change significantly influenced by negatively influenced

**Chl a fluorescence (Lines 12-13):** I would rewrite this paragraph in a more
direct style. For example, it could be rewritten as follow: "*Regarding to the individual*
*effects of both factors on PSIImax, it was found that whereas light conditions decreased*
*it (particularly at HL), OA increased (ca. 17% at LL) or did not affect it (i.e. at ML).*
*Noticeably, lowest values were found under the LightxOA interaction (ca. 0.35) (Fig.*
*2A)*".

**Chl a fluorescence (Lines 20-21):** This paragraph is a little bit confusing. To
avoid any misunderstanding I would rephrase this section as follow: "*Fv/Fm recovery*
*was not influenced neither by light nor pCO2 treatment, excepting under ambient pCO2*
*and ML treatments in which the quantum yield's recovery was maxima (Fig. 2A)*".

**Chl a fluorescence (Lines 22-25):** Similarly than mentioned in the previous
paragraphs, such idea could be simplified as: "*For [RCII], increasing light conditions*
*reduced a 39% the functional RC; however, OA did not show a clear response pattern,*
*as it both decreased (~XX% at LL) as increased (~44% at ML) the functional RC (Fig.*
*3A)".*

**Chl a fluorescence (Lines 26-34):** In my humble opinion, in this paragraph is difficult extracting the main finding, and what is the effect of light, OA and their interaction. Also, I sincerely think that authors should present this dataset at the same way than for figure 1-3, in a bar chart. If I see the table 3 presented, I would summarize the findings as follow: "*In relation to the individual effects of light and OA on the PSII photophysiology it was found that increasing light conditions exerted a mostly stimulatory effect, excepting on connectivity (i.e. P) where it was inhibitory. OA had a significant positive effect at LL on σPSII, ETRmax and Ik, but significantly inhibitory on α at ML. Light and OA, as a single factors, did not exert any significant effect on P and τQa. At the same way than mentioned above, the LightxOA interaction had a contrasting effect on the PSII-photosphysiology, being stimulatory on σPSII, ETRmax and Ik, and inhibitory on P and τQa. The interactive effect of both factors did not alter α*".

**Chl a fluorescence (Lines 37):** I suggest that figure 4 could be moved to supplementary information, as the core information related with this section/variable is contained in Table 3. Thus, and similarly than mentioned above to authors, Table 3 should be presented as figure instead table. Regarding to the Fig.4, authors can highlight into the text, the stimulatory effect of OA on ETR under LL, and the absence of effect by OA under ML. Also, that such stimulatory effect was coupled with significant? lower NPQ values (i.e. LL) or with not significant differences between amb and OA treatments (i.e. ML). Maxima ETR and NPQ values were found under HL.

Regarding to the figure 5, I suggest that it could be also moved to supplementary information together with figure 4, and merge both figures into a single one but with 6 panels.

Finally, despite the main goal of this study was assess the interactive effect of HL and OA, authors did not explicitly quantified such impacts on the response variables assessed. Considering that only the OA effect could be calculated, as cells did not grow under HL, I suggest authors calculating the size effect in percentage (or the log response ratio) at least for the HLxOA interaction and for all variables, and show them together in a final figure. Using this approach could strengthen the message of your work, and easily showing to a potential reader if the HLxOA interaction was synergistic/antagonistic (or stimulatory/inhibitory), and on what variable(s).

**Discussion:** What is the main finding of your study?. It would be nice highlighting to a potential reader in a sentence or a paragraph what is the gap that this study fills (or contributes).

**Lines 28-29:** Based on the fact that significant higher ETRmax, Ik and alpha values were not coupled with higher POC, and ultimately growth, could it be plausible that cells were less efficient than under LL? Or more damaged?. Thus, this lower photosynthetic efficiency (or higher damage) would be consistent with a 2-fold increase in the NPQ, and ultimately, could support the higher Fv/Fm recovery (%).

**Line 34:** I think that "high light" should be modified to avoid any misunderstanding with the HL treatment.

**Line 40:** As the inability for growing of the target specie is a surprising result, and previous results have found that such phytoplankton group grow under HL and stratified conditions, I suggest that authors discuss the potential mechanism(s) or reason(s) that impeded that cell grown.

**Lines 46 and 69:** Statistical information can be omitted in discussion section

**Implications:** On one hand, it is true that the interaction HLxOA stimulated all assessed processes in *G. cryophila*, however, such responses were based on a short-term scale. It could be plausible that such beneficial effects would be accentuated at long-term scales by an adaptation of the populations, or by contrast, reduced. Authors could briefly discuss such issue in the context of their study.

On the other hand, in my humble opinion, I think that this idea presented at the end of the manuscript should be also highlighted into the Introduction section, as it represents the core about why assess the interactive effects of both global-change factors on Antarctic phytoplankton communities. Currently, we do not know how communities will respond to such changes, if such changes will lead to a diatoms or flagellates-dominated community, and as a consequence, if it will boost or reduce the C-sink capacity of this area.

**Minor comments:**

**Tables 1-3:** Please, include what lowercase letters mean. Are they representing posthoc comparisons?. Data represent means and SD?. And OA?.

**Introduction (line 54):** Change almost nothing by little is known

---

## Referee Comment (RC1) · John Beardall (Referee) · 16 Apr 2019

This is an excellent and novel contribution reporting on experiments of the interaction between light intensity and ocean acidification on aspects of the growth and physiology of an Antarctic cryptophyte. Sine information on Southern Ocean cryptophytes is limited, especially with regard to the effects of elevated $CO_2$, the current contribution is especially welcome. The main take home message is that elevated $CO_2$ allows growth at high light, which under present day $CO_2$ levels would be inhibitory (non-permissive) to growth i.e. increased $CO_2$ alleviates photoinhibition.

The data presented are thorough and well-presented. The reasoning put forward to

explain the data is carefully thought through and although I was surprised that the effects of OA were not apparent under medium high levels, I was persuaded by the arguments of the authors that this might be related to costs of increased N metabolism under those conditions.

I have no major criticisms to offer. One minor point is on the last line of the results (page 7 line 9) - where the authors state "...much higher NPQ values were determined in the ambient pCO2 relative to the OA treatment." should this be followed by "in the LL treatment"?

---

## Author Comment (AC1) · 6 May 2019

Our answers to the comments by reviewer #1 are attached as a supplement.

Please also note the supplement to this comment:
https://www.biogeosciences-discuss.net/bg-2019-97/bg-2019-97-AC1-supplement.pdf

---

## Author Comment (AC2) · 6 May 2019

Our answers to the comments by reviewer #2 are attached as a supplement.

Please also note the supplement to this comment:
https://www.biogeosciences-discuss.net/bg-2019-97/bg-2019-97-AC2-supplement.pdf

[Figure]

**Fig. 1.** Figure 3

[Figure]

**Fig. 2.** Figure 4

---

## Author Comment (AC3) · 6 May 2019

The revised manuscript is attached as supplement.

Please also note the supplement to this comment:
https://www.biogeosciences-discuss.net/bg-2019-97/bg-2019-97-AC3-supplement.pdf

---

## Author Response (AR2)

Response to the referee comments of John Beardall on "Ocean acidification and high irradiance stimulate growth of the Antarctic cryptophyte *Geminigera cryophila*"

**John Beardall (Referee #1)**

This is an excellent and novel contribution reporting on experiments of the interaction between light intensity and ocean acidification on aspects of the growth and physiology of an Antarctic cryptophyte. Sine information on Southern Ocean cryptophytes is limited, especially with regard to the effects of elevated  $CO_2$ , the current contribution is especially welcome. The main take home message is that elevated  $CO_2$  allows growth at high light, which under present day  $CO_2$  levels would be inhibitory (non-permissive) to growth i.e. increased  $CO_2$ alleviates photoinhibition.

The data presented are thorough and well-presented. The reasoning put forward to explain the data is carefully thought through and although I was surprised that the effects of OA were not apparent under medium high levels, I was persuaded by the arguments of the authors that this might be related to costs of increased N metabolism under those conditions. I have no major criticisms to offer.

AUTHORS: We thank the reviewer for the kind words on our manuscript.

One minor point is on the last line of the results (page 7 line 9) - where the authors state "...much higher NPQ values were determined in the ambient  $pCO_2$  relative to the OA treatment." should this be followed by "in the LL treatment"?

AUTHORS: The reviewer is right, this information was missing. Hence, it is now written in the revised manuscript on P7 L23-25: "Much higher NPQ values were determined in the ambient  $pCO_2$  relative to the OA treatment under LL while such  $pCO_2$  effect was absent under ML."

**Response to the referee comments of Marco J. Cabrerizo on "Ocean acidification and high irradiance stimulate growth of the Antarctic cryptophyte *Geminigera cryophila*"**

**Marco J. Cabrerizo (Referee #2)**

Trimborn et al. evaluate in a full-factorial experimental design how the interaction between ocean acidification and high light alter the photophysiology, stoichiometry, production and growth of a model Antarctic Cryptophyte at short-term scales. The topic assessed fits with the Biogeoscience's scope, and is novel as currently information related with the interactive effect of both factors on Antarctic phytoplankton is still scarce. The experiment is well-designed and performed, and the results are appropriate to being published in the Journal.

Below, authors will find a point by point revision with the main issues found over the manuscript, and suggestions which I hope that they find useful.

Respectfully submitted, Marco J. Cabrerizo

AUTHORS: We thank the reviewer for the kind words on our manuscript.

**Title:** I suggest including in the title two variables also quantified by authors: production and photophysiology of the species. On potential title could be: "Ocean acidification and high irradiance stimulate the photophysiology, growth and production in the Antarctic cryptophyte Geminigera cryophila".

AUTHORS: To account for the reviewer's comment, we have changed the title to: "Ocean acidification and high irradiance stimulate the photophysiological fitness, growth and carbon production in the Antarctic cryptophyte *Geminigera cryophila*".

**Introduction:** What are the individual effects of HL and OA on phytoplankton?. I suggest that authors add some general information about the individual effects of light and OA on primary producers, and then they focus such impacts on Antarctic phytoplankton and crytophytes. Through a general view, a potential reader can identify the gaps of knowledge related with the quantification of such impacts on phytoplankton, and the scarcity of experimental studies testing their interaction on this key group.

AUTHORS: According to the reviewers' suggestion we have added information on the individual effect of light and  $CO_2$  availability before pointing out what is known in response to OA and HL for Antarctic phytoplankton. It now reads in the revised manuscript:

P2, L10-13: "Light availability strongly influences the rate of growth and carbon fixation of phytoplankton (Falkowski and Raven 2007). With increasing irradiance, Antarctic phytoplankton species exhibited increased growth and carbon fixation, but only until photosynthesis was saturated (Fiala and Oriol, 1990; Heiden et al., 2016). When exposed to excessive radiation, phytoplankton cells can get photoinhibited and even damaged."

P2, L25-30: "For various Antarctic diatoms and the prymnesiophyte *P. antarctica*, growth and/or carbon fixation remained unaltered by OA alone (Riebesell et al., 1993; Boelen et al., 2011; Hoogstraten et al., 2012; Trimborn et al., 2013; Hoppe et al., 2015; Heiden et al., 2016). Recent studies suggest that Southern Ocean diatoms are more prone to OA especially in conjunction with high light than the prymnesiophyte *Phaeocystis antarctica* (Feng et al., 2010; Trimborn et al., 2017a,b; Beszteri et al. 2018; Heiden et al., 2018; Koch et al., 2018, Heiden et al., 2019)."

**Lines 27-29:* If it was found a contrasted response pattern in southern WAP, it means that there were both a positive as a negative effect. In such case, it partially agrees with those results reported in the northern part. Please rewrite.**

AUTHORS: In fact, a contrasted response in chlorophyll a biomass and primary production between the northern and the southern WAP regions was observed. To point this out more clearly, the effects observed in the northern and southern WAP waters are described more clearly to avoid any misunderstanding in this respect on P1, L25-36: "Rising air temperature resulted in shorter sea ice seasons (Smith and Stammerjohn, 2001) with contrasting effects on phytoplankton biomass, composition and productivity between the northern and southern WAP. For the latter, the earlier retreat of sea ice together with the observed increase in surface water temperature led to shallow water column stratification, which favored phytoplankton growth and productivity. In the northern part of the WAP on the other hand, the earlier disappearance of sea ice was associated to greater wind activities and more cloud formation. As a consequence, a deepening of the upper mixed layer was found, providing less favorable light conditions. Next to reduced chlorophyll a accumulation (Montes-Hugo et al., 2009) and primary production (Moreau et al., 2015), a decline of large phytoplankton such as diatoms relative to the whole community was observed (Montes-Hugo et al., 2009; Rozema et al., 2017). Accordingly, a recurrent shift from diatoms to cryptophytes and small flagellates was reported for waters north of the WAP, with important implications for food web dynamics (Moline et al., 2004; Ducklow et al., 2007; Montes-Hugo et al., 2009; Mendes et al., 2017)."

**Lines 29-30: Break this sentence to separate both ideas.* AUTHORS: This section has been rewritten on P1 L25-36.**

**Line 45: Reference?**

AUTHORS: To account for the reviewer's comment the reference Falkwoski and Raven (2007) has been added on P2 L10.

*Line 69:* I think that it would be nice that authors include a hypothesis work about what do they expect based on the previous information known?.

AUTHORS: As suggested by the reviewer, we have included a hypothesis about what we expect based on previous information, it now reads on P2 L36-37: "Based on previous studies on the single effects of light or  $CO_2$  alone, we hypothesize that cryptophytes are able to cope well with OA and high light conditions. Due to the limited information available on Antarctic cryptophyte physiology, this study assessed...".

**Material and Methods:**

**Culture conditions:** This subsection is confusing at the present state. I propose you modifying it as follow: "Before to being used in experimentation (two weeks), triplicate semi-continuous cultures of the Antarctic cryoptophyte G. cryophyla (CCMP 2564) were grown (and maintained in mid-exponential growth) at 2°C in sterile -filtered (0.2 um) Antarctic seawater (salinity 30.03) enriched with phosphate (final concentration 100 umol L-1) and nitrate (final concentration 6.25 umol L-1) (N:P ratio of 16:1, Redfield 1963), as well as, trace metals and vitamins according to F/2 medium (Ref). Cells were exposed under a 16h L: 8h D cycle and three constant light intensities (20, 200 and 500  $\mu$ mol m-2 s-1) (LL, ML and HL, respectively)".

Once you have already described the pre-acclimation phase, you can describe your experimental setup, and finally, all the details related with the  $CO_2$  enrichment. For example, "To assess the interactive effect of light and OA on the photophysiology, metabolism and growth of G. cryophila, a  $3x^2$  full factorial matrix (in triplicate) was implemented with: (a) three light levels (LL, ML and HL) and two p $CO_2$  treatments (ambient and OA)"...

AUTHORS: We have integrated most of the suggested changes by the reviewer in the section dealing with the culture conditions, it now reads:

P3, L5-10: "Triplicate semi-continuous cultures of the Antarctic cryptophyte *Geminigera cryophila* (CCMP 2564) were grown in exponential phase at 2 °C in sterile-filtered (0.2  $\mu$ m) Antarctic seawater (salinity 30.03). This seawater was enriched with phosphate (final concentration of 100  $\mu$ mol L-1), nitrate (final concentration of 6.25  $\mu$ mol L-1) (N:P ratio of 16:1, Redfield, 1963) as well as trace metals and vitamins according to F/2 medium (Guillard and Ryther, 1962). *G. cryophila* cells were grown under a 16h light: 8h dark cycle at three constant light intensities (LL = 20, ML = 200 and HL = 500  $\mu$ mol photons m-2 s-1)..."

P3, L12-17: The three light treatments were further combined with two CO2 partial pressures (pCO2) of 400 (ambient pCO2 treatment) or 1000  $\mu$ atm (OA treatment, Table 1). All pCO2 treatments and the respective dilution media were continuously and gently bubbled through a frit with humidified air of the two pCO2 levels, which were generated from CO2-free air (< 1 ppmv CO2; Dominick Hunter, Kaarst, Germany) and pure CO2 (Air Liquide Deutschland ltd., Düsseldorf, Germany) with a gas flow controller (CGM 2000, MCZ Umwelttechnik, Bad Nauheim, Germany).

P3, L21-22: "G. cryophila cells were acclimated to the matrix of three light intensities (LL = 20, ML = 200 and HL = 500  $\mu$ mol photons m-2 s-1) and two pCO2 levels (ambient = 400 and OA = 1000  $\mu$ atm)..."

**Line 15: Cells in exponential or mid-exponential phase. Please clarify.* AUTHORS: Cells were in exponential growth phase, this has been clarified on P3 L24.**

Statistics: Were ANOVA assumptions checked?. Was the interaction light and OA significant in all response variables? Please clarify. Over the results section, I cannot see if Light×OA was or not significant.

AUTHORS: Of course the ANOVA assumptions were checked. Further, we have added the requested information *throughout* the results section and point now out more clearly whether the interaction of  $CO_2$  and light had a significant influence.

**Results:** I propose authors to change the ordering of the results. First, they should show PSII variables, then pigments, metabolism and stoichiometry and finally growth. The rationale behind my proposal is related with the fact that changes at PSII level occurs in temporal scales ranging mseg to minutes, whereas those related with the other variables need more time.

AUTHORS: Please note that the discussion section mainly focuses on the ecophysiological response und thus concentrates first on the  $CO_2$ -Light responses on growth, elemental stoichiometry and composition and then relates these results to the underlying physiological mechanism such as electron transport rates (ETRs) and their influence on the Calvin cycle through comparison of the ETRs with the POC/N quotas and considers photoacclimation responses incl. chlorophyll a fluorescence characteristics and pigment contents. Hence, the order of the presented results is linked to the order of their presentation in the discussion. As this order is also reflecting the temporal scales as suggested by the reviewer, but rather starting from the long-term towards the short-term responses, we would like to keep the order of the results section as it is.

*Line 30:* In my humble opinion, I think that authors should consider the effect of light under ambient pCO2. Authors state that "growth rates remained unchanged in cells grown under ambient pCO2", however, according to the figure 1, increasing light had a negative effect on cells. In fact, they did not grow.

AUTHORS: The reviewer is completely right and we agree that this negative HL-effect on growth under ambient pCO2 needs to be mentioned. Accordingly, we have modified this section on P6 L6-9: "In response to increasing irradiance, growth rates of cells grown under ambient pCO2 remained unchanged between LL and ML, but were negatively influenced by HL as they were unable to grow. Under OA, however, growth rates significantly increased between LL and ML by 89% (posthoc: p < 0.05) and between ML and HL by 32% (posthoc: p < 0.05), respectively."

*Line 38: Change significantly influenced by negatively influenced* AUTHORS: Agreed and done.

**Chl a fluorescence (Lines 12-13):** I would rewrite this paragraph in a more direct style. For example, it could be rewritten as follow: "Regarding to the individual effects of both factors on PSIImax, it was found that whereas light conditions decreased it (particularly at HL), OA increased (ca. 17% at LL) or did not affect it (i.e. at ML). Noticeably, lowest values were found under the LightxOA interaction (ca. 0.35) (Fig. 63 2A)".

AUTHORS: As suggested, we have rewritten this part and it now reads on P6 L29-32: "The dark-adapted maximum quantum yield of PSII ( $F_v/F_m$ ) was strongly influenced by irradiance (2-way ANOVA: p < 0.0001) and CO2 (2-way ANOVA: p = 0.0012) and their interaction (2-way ANOVA: p < 0.05; Fig. 2A). With increasing irradiance  $F_v/F_m$  generally declined whereas OA increased it at LL (17%, posthoc: p < 0.01) or did not change it at ML. Noticeably, the interaction of HL and OA resulted in the lowest  $F_v/F_m$  value."

*Chl a fluorescence (Lines 20-21):* This paragraph is a little bit confusing. To avoid any misunderstanding I would rephrase this section as follow: "Fv/Fm recovery was not influenced neither by light nor pCO2 treatment, excepting under ambient pCO2 and ML treatments in which the quantum yield's recovery was maxima (Fig. 2A)".

AUTHORS: We agree with the reviewer that this section can evoke easily misunderstanding. Considering that increasing light indeed significantly altered  $F_{\nu}/F_m$  recovery (2-way ANOVA: p < 0.01), it is now written on P6 L34-36: "Neither high pCO2 nor the interaction of light and CO2 affected  $F_{\nu}/F_m$  recovery whereas the increase in irradiance had a significant effect (2-way ANOVA: p < 0.01, Fig. 2B), being increased by 11% between LL and ML under ambient pCO2 (posthoc: p < 0.05)."

**Chl a fluorescence (Lines 22-25):** Similarly than mentioned in the previous paragraphs, such idea could be simplified as: "For [RCII], increasing light conditions reduced a 39% the functional RC; however, OA did not show a clear response pattern, as it both decreased (~XX% at LL) as increased (~44% at ML) the functional RC (Fig. 3A)".

AUTHORS: To simplify this section, it is now written in the revised manuscript on P6 L37-P7 L1: "The increase of CO2 or light alone had no effect on cellular concentrations of functional photosystem II reaction centers ([RCII]) while the interaction of both factors strongly altered [RCII] (2-way ANOVA: p < 0.0001; Fig. 3). From LL to ML [RCII] decreased under ambient pCO2 (39%, posthoc: p < 0.001) while the combination of ML with OA synergistically increased it (44%, posthoc: p < 0.01, Fig. 3). [RCII] was reduced by OA at LL (37%, posthoc: p < 0.01) whereas the combined effect of OA and ML led to an increase (49%, posthoc: p < 0.01)."

Chl a fluorescence (Lines 26-34): In my humble opinion, in this paragraph is difficult extracting the main finding, and what is the effect of light, OA and their interaction. Also, I sincerely think that authors should present this dataset at the same way than for figure 1-3, in a bar chart. If I see the table 3 presented, I would summarize the findings as follow: "In relation to the individual effects of light and OA on the PSII photophysiology it was found that increasing light conditions exerted a mostly stimulatory effect, excepting on connectivity (i.e. P) where it was inhibitory. OA had a significant positive effect at LL on  $\sigma$ PSII, ETRmax and Ik, but significantly inhibitory on  $\alpha$  at ML. Light and OA, as a single factors, did not exert any significant effect on P and  $\tau$ Qa. At the same way than mentioned above, the LightxOA interaction had a contrasting effect on the PSII-photosphysiology, being stimulatory on  $\sigma$ PSII, ETRmax and Ik, and inhibitory on P and  $\tau$ Qa. The interactive effect of both factors did not alter  $\alpha$ ".

AUTHORS: As suggested by the reviewer, throughout the whole result section we have added the requested information for significant effects by light,  $CO_2$  and their combination for each parameter. Furthermore, we have put together a table summarizing the statistical outcome from the two-way ANOVA, which tested for significant differences in response to light,  $CO_2$  and their combination:

|                        | Significant  | Significant            | Significant                   |
|------------------------|--------------|------------------------|-------------------------------|
| Parameter              | light effect | CO 2 effect | light/CO 2 -Effect |
| μ                      | p < 0.01     |                        |                               |
| POC content            |              |                        |                               |
| POC production         | p < 0.01     |                        |                               |
| C:N ratio              |              |                        | p < 0.01                      |
| $F_{\nu}/F_m$          | p < 0.0001   | p < 0.01               | p < 0.05                      |
| $F_{\nu}/F_m$ recovery | p < 0.01     |                        |                               |
| Chl a                  | p < 0.0001   |                        |                               |
| Chl c 2     | p < 0.0001   | p < 0.05               |                               |
| Allo                   | p < 0.0001   | p < 0.01               | p < 0.01                      |
| [RCII]                 |              |                        | p < 0.0001                    |
| Р                      | p < 0.05     |                        |                               |

| σ PSII  |          |          | p < 0.05 |
|--------------------|----------|----------|----------|
| $	au_{Qa}$         | p < 0.05 |          |          |
| ETR max |          | p < 0.05 |          |
| IK                 |          | p < 0.05 |          |
| α                  | p < 0.01 |          | p < 0.01 |

We agree with the reviewer that the overall effects on photophysiology (P,  $\sigma_{PSII}$ ,  $\tau_{Qa}$ , ETRmax,  $I_{\rm K}$  and  $\alpha$ ) by CO2, light and their combination should be presented in a less confusing way and more clearly. Considering, however, that no simple and overarching effect was found that applies for all factors (light,  $CO_2$ ,  $CO_2$ -light, please see table above), we prefer to keep presenting each parameter individually instead of describing them together as proposed by the reviewer, which we belief could create confusion, too. To, however, account for the reviewer's point, which is definitely justified, we instead tried to be more clear and to simplify the description of the photophysiological results (P,  $\sigma_{PSII}$ ,  $\tau_{Oa}$ , ETRmax,  $I_K$  and  $\alpha$ ). It now reads on P7 L1-11: "While CO2 and the interaction of CO2 and light together did not change the energy transfer between PSII units (i.e. connectivity, P), only the increase in irradiance had a significant effect (2-way ANOVA: p < 0.05), reducing P by 22% between LL and ML under OA (posthoc: p < 0.05, Figure 3B). While the increase in CO2 or light did not alter the functional absorption cross-sections of PSII ( $\sigma_{PSII}$ ), the interaction of both factors, however, had an effect (2-way ANOVA: p < 0.05; Figure 3C).  $\sigma_{PSII}$  values were similar under LL and ML at ambient pCO2. The interaction of OA and ML, however, lowered them (posthoc: p < 0.05, Table 3). On the other hand, when grown under OA  $\sigma_{PSII}$  was larger under HL than under ML (1-way ANOVA: p < 0.01). Re-oxidation times of the primary electron acceptor  $Q_a$  ( $\tau_{Oa}$ ) significantly changed with increasing irradiance (2-way ANOVA: p < 0.05), but not by high CO2 or the interaction of both factors together (Figure 3D).  $\tau_{Qa}$  of OAacclimated cells was much shorter at HL than at ML (*1-way ANOVA*: p < 0.05)."

P7 L13-19: "Both maximum absolute electron transport rates (ETRmax) and minimum saturating irradiances ( $I_{\rm K}$ ) followed the same trend and were significantly changed by CO2 (2-*way ANOVA*: p < 0.05), but not by light or the interaction of both factors (Table 3). OA significantly enhanced both parameters under LL (ETRmax: posthoc: p < 0.05,  $I_{\rm K}$ : posthoc: p < 0.05), but not under ML. The maximum light utilization efficiency ( $\alpha$ ) was significantly affected by light (2-*way ANOVA*: p < 0.01) and the interaction of CO2 and light (2-*way ANOVA*: p < 0.01), but not by CO2 alone (Table 3).  $\alpha$  significantly increased from LL to ML at ambient pCO2 (53%, posthoc: p < 0.01) while such effect was absent under ML and OA. Between ML and HL,  $\alpha$  did not differ when grown under OA."

As proposed by reviewer the parameters [RCII], *P*,  $\sigma_{PSII}$  and  $\tau_{Qa}$  are now presented in the same figure showing bar charts. Please see Figure 3.

**Chl a fluorescence (Lines 37):** I suggest that figure 4 could be moved to supplementary information, as the core information related with this section/variable is contained in Table 3. Thus, and similarly than mentioned above to authors, Table 3 should be presented as figure instead table. Regarding to the Fig.4, authors can highlight into the text, the stimulatory effect of OA on ETR under LL, and the absence of effect by OA under ML. Also, that such stimulatory effect was coupled with significant? lower NPQ values (i.e. LL) or with not significant differences between amb and OA treatments (i.e. ML). Maxima ETR and NPQ values were found under HL.

Regarding to the figure 5, I suggest that it could be also moved to supplementary information together with figure 4, and merge both figures into a single one but with 6 panels.

Finally, despite the main goal of this study was assess the interactive effect of HL and OA, authors did not explicitly quantified such impacts on the response variables assessed. Considering that only the OA effect could be calculated, as cells did not grow under HL, I

suggest authors calculating the size effect in percentage (or the log response ratio) at least for the HLxOA interaction and for all variables, and show them together in a final figure. Using this approach could strengthen the message of your work, and easily showing to a potential reader if the HLxOA interaction was synergistic/antagonistic (or stimulatory/inhibitory), and on what variable(s).

AUTHORS: The reviewer suggested to "move figure 4 to supplementary information, as the core information related with this section/variable is contained in Table 3". As in our opinion it is important to show on which basis the curve fitting results (shown in Table 3) were derived, we find it indeed important to show the shape of the PI curves and whether the fitting fits well to the curves. Hence, we would like to keep Figure 4 in the main manuscript.

Further the reviewer asks to include *P*,  $\sigma_{PSII}$  and  $\tau_{Qa}$  into figure 3, this was done. Please see Figure 3.

The reviewer also proposed to highlight in the text the stimulatory effect of OA on ETR and NPQ under LL and the absence of effect by OA under ML". This was done and it is now written on P7 L15-16: "OA significantly enhanced both parameters under LL (ETRmax: posthoc: p < 0.05,  $I_{\rm K}$ : posthoc: p < 0.05), but not under ML."

On P7 L22-24: "Much higher NPQ values were determined in the ambient  $pCO_2$  relative to the OA treatment under LL while such  $pCO_2$  effect was absent under ML"

Moreover, the reviewer suggests to move figure 5 to the supplementary material and also to combine Figure 4 and 5. As suggested we combined Figure 4 and 5 into one figure, which is now Figure 4, but refrain from moving this figure into the supplementary material as the content of this figure is critical for the discussion of the whole data set.

As suggested by the reviewer we have calculated the log response ratio (RR) for the OA HL treatment relative to the ambient  $pCO_2$  ML treatment according to Lajeunesse (2015, *Ecology* 96: 2056-2063). The obtained results are shown in the following table:

| Parameter              | RR    |
|------------------------|-------|
| μ                      | 0.41  |
| POC content            | -0.07 |
| POC production         | 0.33  |
| C:N ratio              | -0.12 |
| $F_{\nu}/F_m$          | -0.16 |
| $F_{v}/F_{m}$ recovery | -0.01 |
| Chl a                  | 0.08  |
| Chl c 2     | -0.12 |
| Allo                   | 0.28  |
| [RCII]                 | -0.20 |
| Р                      | -0.22 |
| σ PSII      | 0.17  |
| $	au_{Qa}$             | -0.12 |
| ETR max     | 0.33  |
| IK                     | 0.14  |
| α                      | 0.19  |

It is true that a positive RR was found for growth and POC production for the OA HL treatment. Considering, however, that negative responses were observed for instance for  $\tau_{Qa}$ , this does not necessarily mean a negative effect *per se* on phytoplankton physiology. In fact ETRmax,  $I_k$  and  $\alpha$  were positive. On the other hand, re-oxidation times of Qa were shorter and thus indicating faster electron drainage into downstream processes. Hence, a negative RR value does not necessarily indicate a negative response in phytoplankton physiology. Due to

the latter finding and the potential for misinterpretation of the results, we refrain from adding this table to the manuscript.

**Discussion:** What is the main finding of your study? It would be nice highlighting to a potential reader in a sentence or a paragraph what is the gap that this study fills (or contributes).

AUTHORS: As suggested by the reviewer, we have added the following at the beginning of the discussion on P7 L26-29: "Ecophysiological studies on Antarctic cryptophytes to assess whether climatic changes such as ocean acidification and enhanced stratification affect their growth in Antarctic coastal waters in the future are lacking so far. This study can show that the Antarctic cryptophyte *G. cryophila* may be a potential winner of such climatic conditions as it reached highest rates of growth and particulate organic carbon production when grown under HL and OA."

**Lines 28-29:** Based on the fact that significant higher ETRmax, Ik and alpha values were not coupled with higher POC, and ultimately growth, could it be plausible that cells were less efficient than under LL? Or more damaged?. Thus, this lower photosynthetic efficiency (or higher damage) would be consistent with a 2-fold increase in the NPQ, and ultimately, could support the higher Fv/Fm recovery (%).

AUTHORS: Please note that  $\text{ETR}_{\text{max}}$  and POC production are not necessarily linked by a 1:1 ratio. Linear electron transport is directly coupled to the Calvin cycle, nitrite reduction and other electron accepting processes. In our data set, POC production remained the same between LL and ML under ambient pCO2, this indicates saturation of the Calvin cycle and thus induction of alternative electron transport pathways. ETRs on the hand do not only reflect linear electron flow, but also comprise alternative electron pathways. Therefore the increase in ETRmax between LL and ML under ambient pCO2 did not yield higher POC production and rather points towards alternative electron transport pathways. Hence, cells photosynthesize as efficiently under LL as under ML, but additionally at ML more electrons are fed into alternative pathways such as xanthophyll cycling as observed by the increase in NPQ. In fact, cells are not damaged, but rather are characterized by a higher  $F_v/F_m$  recovery, supporting a lack of damage. Due to the latter observations, we would like to keep this part of the discussion as it is.

*Line 34:* I think that "high light" should be modified to avoid any misunderstanding with the *HL treatment.*

AUTHORS: To be more clear, it is now written on P8 L15: "Unexpectedly, *G. cryophila* was, however, unable to grow at 500  $\mu$ mol photons m-2 s-1...".

*Line 40:* As the inability for growing of the target specie is a surprising result, and previous results have found that such phytoplankton group grow under HL and stratified conditions, I suggest that authors discuss the potential mechanism(s) or reason(s) that impeded that cell grown.

AUTHORS: We agree with the reviewer and therefore have now written on P8 L18-21: "A connection of this group with high illuminated conditions was first suggested by Mendes et al. (2017), but lacks information which cryptophyte species were present and their photosynthetic responses. The reason for this difference could be related to species- or strain-specific differences."

*Lines 46 and 69: Statistical information can be omitted in discussion section* AUTHORS: Done as suggested.

**Implications:** On one hand, it is true that the interaction HLxOA stimulated all assessed processes in G. cryophila, however, such responses were based on a short-term scale. It could be plausible that such beneficial effects would be accentuated at long-term scales by an adaptation of the populations, or by contrast, reduced. Authors could briefly discuss such issue in the context of their study.

On the other hand, in my humble opinion, I think that this idea presented at the end of the manuscript should be also highlighted into the Introduction section, as it represents the core about why assess the interactive effects of both global-change factors on Antarctic phytoplankton communities. Currently, we do not know how communities will respond to such changes, if such changes will lead to a diatoms or flagellates-dominated community, and as a consequence, if it will boost or reduce the C-sink capacity of this area.

AUTHORS: The reviewer points out that the observed responses were obtained through performance of a short-term experiment, while responses could be different on the longer term. To account for this point, we have added the following sentence in the Implications section on P9 L35-38: "Our results from a short-term  $CO_2$ -light experiment point towards a high ability of *G. cryophila* to acclimate to such conditions and to cope well with medium, but not high irradiances, whether this applies for other Antarctic cryptophyte species as well needs further testing. Also it remains unclear whether similar responses would be found when exposed on a longer term."

The other suggestion by the reviewer was to point out already in the introduction the implications on carbon biogeochemistry from a shift from diatoms to flagellates. This has been done on P2 L7-9: "Hence, higher abundances of cryptophytes could have important implications for the biogeochemistry of these waters, as they are considered to be inefficient vectors of carbon and thus could reduce the efficiency of the biological carbon pump ".

**Minor comments:**

*Tables 1-3:* Please, include what lowercase letters mean. Are they representing posthoc comparisons? Data represent means and SD? And OA?

AUTHORS: The reviewer is right, the asked information was missing and has been added. Thanks for pointing this out.

**Introduction (line 54): Change almost nothing by little is known**

AUTHORS: Agreed, it now reads on P2 L20-22: "While laboratory studies so far mainly have concentrated to disentangle the physiological response of Southern Ocean key species of diatom and prymnesiophytes to different environmental factors almost nothing is known on Antarctic cryptophytes."